

**Soil organic carbon stocks are systematically overestimated by**
**misuse of the parameters bulk density and stone content**
Christopher Poeplau, Cora Vos, Axel Don
Thünen Institute of Climate-Smart Agriculture, Bundesallee 50, 38116 Braunschweig, Germany
*Correspondence to*: Christopher Poeplau, christopher.poeplau@thuenen.de
Key words: Fine soil stock, soil skeleton, rock content, bulk density estimation


















**Abstract** Estimation of soil organic carbon (SOC) stocks requires estimates of the carbon content, bulk density,
stone content and depth of a respective soil layer. However, different application of these parameters could
introduce a considerable bias. Here, we explain why three out of four frequently applied methods overestimate
SOC stocks. In stone rich soils (>30 Vol. %), SOC stocks could be overestimated by more than 100%, as
revealed by using German Agricultural Soil Inventory data. Due to relatively low stone content, the mean
systematic overestimation for German agricultural soils was 2.1-10.1% for three different commonly used
equations. The equation ensemble as re-formulated here might help to unify SOC stock determination and avoid
overestimation in future studies.

## 1 Introduction

Size and changes in the soil organic carbon (SOC) pool are major uncertainties in global earth system models
used for climate predictions. Accurate estimation of SOC stocks is vital to understanding the links between
atmospheric and terrestrial carbon (Friedlingstein et al., 2014). Estimates of global SOC stocks are based on soil
inventories from regional to continental scale, involving multiplication of measured carbon content by soil bulk
density (BD, oven-dry mass of soil per unit volume) and the depth of the respective soil layer (Batjes, 1996). The
content of elements such as carbon and nitrogen in soils is usually determined in an aliquot sample of the fine
soil, which is defined as the part of the soil that passes through a 2 mm sieve (Corti et al., 1998). Mineral
fragments >2mm (gravel and stones) are considered free of SOC (Perruchoud et al., 2000), although this may not
be completely true as shown by Corti et al. (2002). Furthermore, living root fragments >2mm are not considered
part of SOC, but usually as part of plant biomass. It is thus widely accepted that accurate estimates of SOC
stocks should account in some way for the presence of gravel, stones (coarse soil, in the following only referred
to as stones) and roots (Rytter, 2012; Throop et al., 2012).
The accuracy of SOC estimates depends in the first instance on the available data and their quality. Soil organic
carbon content of the fine soil is usually measured with high throughput and precision in elemental analyzers,
while BD and stone content are often only assessed in plot scale studies due to much more elaborate sampling
requirements (Don et al., 2007). In regional scale studies or national soil inventories, BD is therefore often
approximated using pedotransfer functions and the fraction of stones is often ignored (Wiesmeier et al., 2012).
Stoniness is therefore regarded as the greatest uncertainty in SOC stock estimates (IPCC, 2003). However, even
when all parameters are recorded, considerable difference in SOC stocks can arise from varying use of the
parameters in equations. Apart from the methodological bias caused by using different methods for determining
BD and stone content (Beem-Miller et al., 2016; Blake, 1965), the different calculation approaches could lead to
systematically different SOC stock estimates if soils contain stones. Several of the approaches commonly used to
calculate SOC stocks are not correct and inflate SOC stocks. The aim of this study was i) to reveal the
conceptual differences in widely used methods for SOC stock calculation, ii) to quantify the methodological bias
in SOC stocks in a regional scale soil inventory, iii) to identify the most affected soil layers and finally iv) to
suggest the most adequate method for  unified and unbiased SOC stock calculation.



## 2 Materials and Methods

In a preliminary literature review we selected a total of 100 publications for which the method used to calculate SOC stocks was recorded. The search was restricted to publications listed in ISI Web of Knowledge, where 'soil carbon stocks' was used as the search term. We ordered the 4915 search results by 'relevance', excluded reviews and modeling studies and avoided redundant senior authors (Tab. S1). In the literature we identified four different methods where identified, which vary in use of the parameters BD and stone content (Henkner et al., 2016; Lozano-García and Parras-Alcántara, 2013; Poeplau and Don, 2013; Wang and Dalal, 2006):

In method one (M1), a certain volume of soil is sampled, dried and weighed to determine BD. Thereby, no separation into fine soil and coarse soil (gravel, stones, roots) fraction is made, while C concentration is determined in a sieved fine soil sample (usually <2 mm). Soil organic carbon stocks are then calculated as follows:

M1:

$$BD_{sample} = \frac{mass_{sample}}{volume_{sample}} \qquad \text{(Eq. 1)}$$

$$SOCstock_i = SOCcon_{fine\ soil} \times BD_{sample} \times depth_i \qquad \text{(Eq. 2)}$$

where $BD_{sample}$ is the bulk density of the total sample, $SOCstock_i$ is the SOC stock of the investigated soil layer (i) [Mg ha$^{-1}$], $SOCcon_{fine\ soil}$ is the content of SOC in the fine soil [%] and $depth_i$ is the depth of the respective soil layer [cm]. This method leads to biased SOC stocks estimates due to inadequate representation of the stones as almost SOC free mass. In method two (M2), a certain volume of soil is sampled, dried and weighed. However, after sieving, the mass and volume of stones and coarse roots are determined. In the following, we simplify the equations by omitting coarse roots, which is also 'common practice', although the volume occupied by roots can be considerably high. This source of error is not further discussed in this study. By approximating a stone density ($\rho_{stones}$) of 2.6 g cm$^{-3}$ (root density is usually assumed to be close to 1 g cm$^{-3}$), BD of the fine soil is subsequently calculated as:

M2:

$$BD_{fine\ soil} = \frac{mass_{sample} - mass_{stones}}{volume_{sample} - \frac{mass_{stones}}{\rho_{stones}}} \qquad \text{(Eq. 3)}$$

$$SOCstock_i = SOCcon_{fine\ soil} \times BD_{fine\ soil} \times depth_i \qquad \text{(Eq. 4)}$$

Thus in M2, coarse soil content is accounted for in equation (3), not in equation (4). The opposite is true for the next method (M3), in which the *stone fraction* [dimensionless] is determined, but only applied to reduce the soil volume (Eq. 6), and not to determine $BD_{fine\ soil}$:

M3:



$BD_{sample} = \frac{mass_{sample}}{volume_{sample}}$      (Eq. 1)
$SOCstock_i = SOCcon_{fine\ soil} \times BD_{sample} \times depth_i \times (1 - stone\ fraction)$      (Eq. 5)
In method four (M4), the coarse soil fraction is accounted for in both equations, i.e. to calculate $BD_{fine\ soil}$ (Eq.
3) and the volume of the fine soil (Eq. 3)
M4:
$BD_{fine\ soil} = \frac{mass_{sample} - mass_{stones}}{volume_{sample} - \frac{mass_{stones}}{\rho_{stones}}}$      (Eq. 3)
$SOCstock_i = SOCcon_{fine\ soil} \times BD_{sample} \times depth_i \times (1 - stone\ fraction)$      (Eq. 5)
In the German Agricultural Soil Inventory, more than 3000 agricultural soils (cropland and grassland) have been
sampled as described by Grüneberg et al. (2014). To date, a total of 2515 sites were sampled and analysed for all
relevant parameters (stone content, fine soil mass, carbon content of the fine soil) in five different depth
increments: 0-10, 10-30, 30-50, 50-70 and 70-100 cm. Here, we considered only mineral soils with a SOC
content <8.7% giving a total of 2350 sites and 11,514 soil samples. We expected the strongest effects in soils
with high stoniness and therefore stratified the dataset by stone content [vol. %]. Statistical analyses were not
conducted, since a systematic deviation implied significant difference between all methods. Data analysis and
plotting was performed in the R 3.1.2 environment (R Development Core Team, 2010).

**3 Results and Discussion**
**3.1 Bias of three calculation methods to estimate SOC stocks**
Three out of the four SOC calculation methods produced systematically overestimated SOC stocks. These
deviations are systematic errors (bias) that cannot be reduced with optimised methods to determine the
parameters SOC content, BD and stone content but reduce the accuracy of SOC stock estimates. As expected,
the differences in SOC stocks between calculation methods increased with stone content (Fig. 1). This is in line
with findings by Rytter (2012) that the method of BD estimation is most important in very stony soils. While
differences between methods for soils with a stone content of less than 5 vol. % were small to almost negligible,
M1-M3 deviated strongly from M4 in soils with >30% stones (Fig. 1, Tab. 1). Since M4 is the closest
approximation to reality, the systematic bias was expressed as relative deviation from M4 (Tab. 1). In soils with
>30% stones, M1 caused the highest bias of all three calculation methods, overestimating SOC stocks by on
average 144%, i.e. more than doubling the real SOC stocks. Methods M2 and M3 also produced biased SOC
stocks with 98% and 21% overestimations for the highest stone content class (>30% stone content). Using an
average $BD_{fine\ soil}$ of 1.2, we plotted the deviation from M4 as a function of volumetric stone content for M1-
M3 (Fig. 2). Thereby, M1 and M2 showed exponential responses, while M3 showed a linear response. These
responses would increase with decreasing bulk density of the fine soil. The literature review revealed that M1,
M2, M3 and M4 were used by 52, 5, 36, and 7 studies respectively. Thus, in 93% of all studies reviewed, SOC



stocks were systematically overestimated assuming a stone fraction >0. More than half of the studies reviewed
did not account for the stone fraction at all.
The number of soils with high stone contents in the German dataset is limited due to the dominance of parent
material from glacio-fluvial deposits (Tab. 1). Thus, the majority of soils (67-78%, depending on soil depth
increment) had a volumetric stone content of <5%. As a consequence, the average SOC stocks were only
moderately influenced by the calculation method (2.1-10.1% deviation, Tab. 1). For forests, which are usually
found on soils less suitable for agriculture, e.g. due to high stoniness, the bias would be stronger. Overall, the
results highlight the importance of a correct use of the parameters BD and stone fraction when calculating SOC
stocks.
**3. 2 Evaluation of the four different calculation methods**
Since all four methods use the same $SOCcon_{fine\ soil}$ due to equal preparation of the fine soil, differences
between the calculation methods arise from differences in use of the parameters BD and stone content. The
individual bias of each method is visualized in Figure 3. In M1, BD of the soil containing SOC (fine soil) is
overestimated due to inclusion of stones in the BD estimate. The mass of fine soil present in the respective soil
layer is also overestimated, since the stone fraction is not subtracted from the soil volume with which $BD_{sample}$
is multiplied (Eq. 2). Thus, M1 'fills' the space occupied by stones with fine soil with an overestimated BD. In
the German Agricultural Soil Inventory, only 9% of all sampled layers were found to be free of stones. Thus, for
most soils M1 is not the correct way to calculate SOC stocks. Similarly, M2 overestimates $SOCstock_i$ by filling
the volume of stones with fine soil. However, BD is calculated and used correctly leading to a smaller systemic
overestimation of SOC compared to M1. Finally, M3 correctly accounts for the stone fraction that can be
assumed to be SOC free. However, in M3 an overestimated BD is applied as in M1, i.e. $BD_{sample}$ and not the
$BD_{fine\ soil}$. Methods to estimate BD and stone content vary, primarily owing to size and abundance of the latter
and may have large uncertainty (Blake, 1965; Parfitt et al., 2010; Rytter, 2012). However, the presented
difference between calculation methods is independent of the method of determination of the these parameters
with one exception: If the sampled soil layer contains no gravel, but only fine soil and stones that exceed the
diameter of a soil ring used to determine $BD_{sample}$, and this ring is placed at a position (in the profile wall)
which is completely free of stones, while the stone content is estimated with a different method and accounted
for, then M3 does resemble M4. Bulk density is often determined with soil rings with a volume between 100 and
500 cm³ or soil probes (Walter et al., 2016). In the German Agricultural Soil Inventory, 250 cm³ soil rings are
used to determine BD. In 89% of all soils inventoried, small stones were detected which end up in the soil ring
and have to be corrected for. Thus, method M3 is rarely a correct method to estimate SC stocks. It is erroneously
often cited as the IPCC default method. However, while the equations given in IPCC resemble M3, IPCC
provides a footnote that is most likely often overlooked, which states that BD estimates should be corrected for
the proportion of 'coarse fragments' (IPCC, 2003). Even if the stone fraction might store a certain amount of
organic carbon (Corti et al., 2002), which might lead to slight underestimation of SOC stocks in M4, we suggest
use of this method in future studies.
**3.3 Proposed equations to calculate SOC stocks**





Bulk density might be of interest as an important soil property. However, for the calculation of SOC stocks alone
it is not needed, while it is the fine soil stock of the investigated soil layer ($FSS_i$, Mg ha$^{-1}$) that is of interest since
it contains the SOC. Thus, the equations in M4 could be reformulated as:
$$FSS_i = \frac{mass_{finesoil}}{volume_{sample}} \times depth_i \qquad \text{(Eq. 6)}$$
$$SOCstock_i = SOCcon_{fine\ soil} \times FSS_i \qquad \text{(Eq. 7)}$$
This has implications for sample preparation: For $BD_{fine\ soil}$ the volume of coarse fragments has to be estimated
by weighing stones and coarse roots separately, while $FSS_i$ would only need the total mass of the fine soil
contained in the known volume of sample. When using the probe method, SOC stock calculation can thus be
simplified, while $FSS_i$ is slightly more complicated to calculate when the profile pit method (determination of
stones within and outside the soil ring) is used (Grüneberg et al., 2014).

**4 Conclusions**
We show here that substantially different methods are used for the calculation of SOC stocks. These methods
differ in use of the parameters bulk density and coarse soil fraction, which causes systematic overestimation of
SOC stocks in three out of four, more or less frequently applied methods, or in 93 of 100 publications reviewed.
We showed that this overestimation can exceed 100% in stone-rich soils. For future studies, we suggest to
calculate the fine soil stock of a certain soil layer which is to be multiplied with its SOC content to derive
unbiased SOC stock estimates.

**Acknowledgements**
This study was funded by the German Federal Ministry of Food and Agriculture in the framework of the German
Agricultural Soil Inventory.












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

**List of Tables**
Table 1: Fraction of total observations for different volumetric stone content classes in the German Agricultural
Soil Inventory and average soil organic carbon stock deviations [%] from M4 for the calculation methods M1-
M3 in different depth increments.

| Depth | Fraction of total observations | | | | | Average relative deviation from M4 | | |
|---|---|---|---|---|---|---|---|---|
| | <5% | 5-10% | 10-20% | 20-30% | >30% | M1 | M2 | M3 |
| 0-10 | 78.4 | 12.9 | 5.7 | 1.8 | 1.2 | 6.1 | 3.6 | 2.2 |
| 10-30 | 72.4 | 14.0 | 6.4 | 3.1 | 4.2 | 7.3 | 4.3 | 2.5 |
| 30-50 | 68.4 | 10.3 | 6.4 | 4.1 | 10.7 | 8.4 | 5.3 | 2.2 |
| 50-70 | 67.5 | 9.4 | 6.4 | 4.1 | 12.6 | 8.8 | 5.8 | 2.1 |
| 70-100 | 68.4 | 9.3 | 5.7 | 3.3 | 13.3 | 10.1 | 6.5 | 2.3 |



















**Figures**

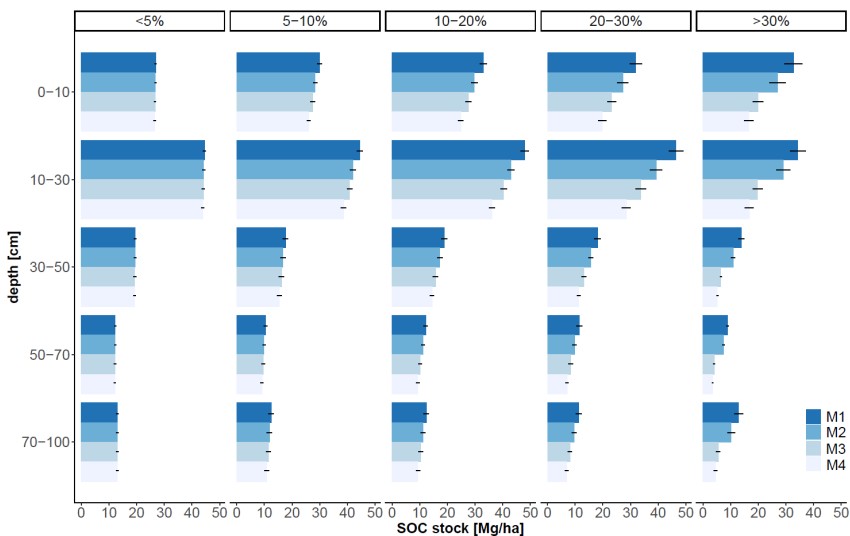


Figure 1: Soil organic carbon stocks of the German Agricultural Soil Inventory in different depth increments
calculated by different calculation methods (M1-M4) for five volumetric stone content classes. Error bars
indicate standard errors.

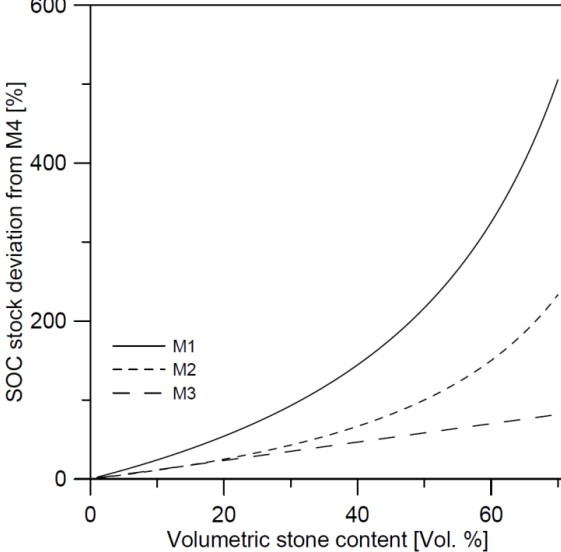


Figure 2: Systematic deviations in SOC stock from calculation method M4 for methods M1-M3 as a function of
volumetric stone content. Bulk density of the fine soil was set to 1.2 [g cm$^{-3}$] in this example.






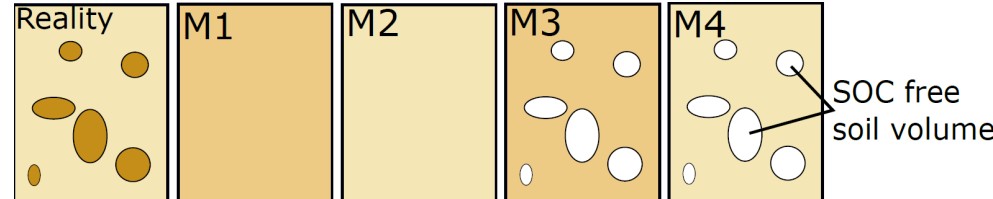


Figure 3: Schematic overview on the four methods applied to estimate the mass of soil needed to calculate soil
organic carbon stocks. Different shades of brown are used to indicate different densities. Thereby the stone
fraction (ellipsoids) has the darkest brown and the fine soil fraction the lightest brown.