# Peer review of "Soil organic carbon stocks are systematically overestimated by misuse of the parameters bulk density and rock fragment content"

_SOIL, 2016_

## Referee Comment (RC1) · Anonymous Referee #1 · 11 Jan 2017

Dear Editor,

In the submitted manuscript, Dr. C. Popleau and co-authors compared four methods to calculate the soil carbon stocks and quite easily demonstrated that three of them overestimated from 2 to 10% the carbon content. Moreover, in stony soils, the overestimation was up to 100%. They used the German Agricultural Soil Inventory dataset to test the calculation method.

GENERAL COMMENTS Since soils represent the largest carbon reservoir of the terrestrial ecosystems, its correct estimation is essential to model the interactions between the pedosphere and the vegetation and predict the effects of climate change on ecosystems. The manuscript addresses an important topic which surely falls within the

scopes of SOIL. The problems highlighted by the authors are not new to the scientific community however, probably for the first time, different methods of soil carbon stock calculation were compared by application to a common dataset therefore allowing to quantify the bias introduced by each of them. The manuscript is well structured, objectives are clear and methods are sound. The results are well supported by the data. I therefore recommend acceptance after some minor corrections are made.

SPECIFIC COMMENTS Add something about soil classification, to which soil types do the soils belong to? Since you have 2350 sites, give at least some general information on more common soil types and parent material. Why did you choose to select the soils with a SOC <8.7%? Or is this this 8.7% the maximum SOC value of the selected plots? Table S1. Try to sort the data and indicate the land use type: Forest, Cropland, Grassland. Based on this classification, you can then derive if the overestimation of soil C stock was prevalent in a certain land use type. Maybe one of the four calculation methods was used more frequently in a certain land use type? Check if this is feasible. lines 107-108: why statistical analyses were not conducted? I'm not sure I understood, rephrase the sentence or explain in a different way. Or delete it if not pertinent with the rest of the manuscript.

TECHNICAL CORRECTIONS *line 70: delete "where identified". *line 79: list the terms in the order they appear in the equation. i.e. SOC stocki, SOC con fine soil, etc. *lines 95-101: I would suggest not to repeat the equations which were already reported in the page before, but to cite them instead. *Figure 1: add a top x axis title "Volumetric stone classes".

---

## Short Comment (SC1) · 12 Jan 2017

Dear Editor, currently, scientists, politicians and environmental managers are desperately seeking for carbon sinks to improve the global carbon budget. Thus, it is important to use appropriate models or to improve models by identifying their shortcomings. Thus, the authors raise a very important issue dealing with the estimation of the soil organic carbon pools, which is the largest terrestrial carbon pool, and show the weakness of four SOC stock equations concerning the bias by comparing and evaluating them. I highly recommend publishing the manuscript. Nevertheless, I have some questions to the authors and some recommendations how the manuscript could be improved.

SPECIFIC COMMENTS Line 51: In line 44, you already mentioned that mineral frag-

ments > 2mm contain pieces of the size classes "gravel" and "stone". But it is not clear how you use the term "stone content" in your manuscript. For example in the Guidelines for soil description of the FAO, the size class of "Coarse gravel" is between 2 and 6 cm and "stones" are classified as mineral pieces between 6 and 20 cm. Can you clarify this? I confirm the comment of Referee #1, that the equations should not be repeated. Maybe they could be summarized in a table, which would allow to repeat the sub-equations in the columns (e.g. column 1: Method, C. 2: BD equation, C. 3: SOC stock equation) Line 70-71: Did these four authors which you cited develop the equations or did they only apply them? In the second case, please refer to the original reference. For example, Henkner et al. (2016) cited four publications concerning the SOC stock equation which was used by them. And during the description of the equations, could you refer to the original references in each section? Line 86: Do you have any references which underline that you use a stone density of 2.6? Or is it the average stone density of the dataset which you used? Line 107: What was the criterion for the classification of the data of the stone content? How did you choose the boundaries of the classes? Lines 118-119: I do not see this observation in Table 1. Line 124: This sentence should be part of the methods. Is BD fine soil of 1.2 the average of the whole dataset (German Agricultural Soil Inventory)? Line 174: Would it be difficult to recalculate SOC stocks which were done by other equations in former times by using Eq 7? Which modification of the raw data is needed? I ask you because such improvements may fail and do not find acceptance due to the bias which is produced by the change of the method. Therefore, it is the simplest way to apply a standardized and established method for dealing with time-series, for example, to keep the comparability at a high level. Could you therefore clarify how data of soil inventories like the inventory which you used must be re-organized first or could be used that the already taken samples and available data is applicable with Eq. 7 in the future? Table 1: In my opinion you show two different topics in this table. Is it therefore meaningful to show them in one table? Table 1 and Figure 1: Is the data of Tab. 1 and Fig. 1 in a way comparable? In my opinion, the highest differences and hence the relative deviation of M4 is found

in sampling depth 10-30 cm and not in 70-100cm (Figure 1). However, in Table 1 the highest value concerning the average relative deviation was found in 70-100 cm. Figure 1: Did you compute any significance test that you can show which method (M1-M3) differs significantly from M4 per depth and stone class? Figure 1: Clarify what kind of values you show. Average values? Figure 2: Would it be possible to show the 95%-confidence intervals of each of the functions? This would show the variability. Figure 2: How did you calculate the systematic deviation? Please, show the formulas at least in the Figure. Can you explain it in the method section? (see above comment line 124). Figure 2: What is the number of replicates per depth and stone class? Is the number of replicates reasonable for the different variabilities, to some extent? Further, could it be that samples of depth horizons 50-70 and 70-100 cm originate from deep soils like "Parabraunerde", "Böden auf Lössderivaten" or "Böden auf Sand", which can contain a low stone content in the upper sampling horizons and maybe also in the deeper sampling horizons? And soil from e.g the "Schwarzwald", "Mittelgebirge" and the Alps do not reach depths of 50, 70 or 100 cm. In these cases, the stone content is already high in the upper sampling layers like 0-10, 10-30 or 30-50 cm. Would this explain the higher variability in 10-30 cm besides the fact that the different depth intervals contributes also to the SOC stock (depthi)? Because during the single stratification by stone content, it seems that the different sampling horizons of all used test sites were mixed. Hence, the stone class (> 30 %) in depths 0-10, 10-30 and 30-50 is represented by sites from rugged terrain while the depth classes 50-70 and 70-100 cm contain samples of "deep soils" which show a stone content > 30 % only deeper than 50 cm soil depth. How does this issue influence your results? Therefore, I agree with Referee #1 comment to re-classify the dataset to avoid such an assumed mixture. For this, the parameter "depth" in the inventory could be the classification factor. Or it could be possible to apply a spatial classification by soil type or region. This would be the first step. After this, the sub-datasets can be stratified by the stone content in a second step. Figure 3: Where do you refer to this figure. In my opinion, it should be Figure 1 and placed in the method section.

TECHNICAL ERRORS Line 158: Do you mean SOC stocks instead of SC stocks?

---

## Author Comment (AC1) · 18 Jan 2017

We are grateful for the comments of the reviewer which were very positive and constructive and hope that we were able to improve the manuscript.

GENERAL COMMENTS Since soils represent the largest carbon reservoir of the terrestrial ecosystems, its correct estimation is essential to model the interactions between the pedosphere and the vegetation and predict the effects of climate change on ecosystems. The manuscript addresses an important topic which surely falls within the scopes of SOIL. The problems highlighted by the authors are not new to the scientific community however, probably for the first time, different methods of soil carbon stock calculation were compared by application to a common dataset therefore allowing to

quantify the bias introduced by each of them. The manuscript is well structured, objectives are clear and methods are sound. The results are well supported by the data. I therefore recommend acceptance after some minor corrections are made.

SPECIFIC COMMENTS Add something about soil classification, to which soil types do the soils belong to? Since you have 2350 sites, give at least some general information on more common soil types and parent material.

Response: We added the following sentence: "The most common soil types sampled were cambisols (24 %), anthrosols (16 %), stagnosols (13%) and albeluvisols (11 %) and the parent material was at 93 % of all sites loose sediments of varying origins."

Why did you choose to select the soils with a SOC <8.7%? Or is this this 8.7% the maximum SOC value of the selected plots?

Response: 8.7% SOC (or 15% SOM) is in the German soil classification the upper boarder for pure mineral soils. Organic soils, or such in transition between organic and mineral soils were excluded. We now added a citation in the respective sentence. We changed the sentence as follows: "Here, we excluded soils with a SOC content >8.7%, which are not considered mineral soils anymore (Ad-Hoc-Ag Boden, 2005), giving a total of 2350 sites and 11,514 soil samples."

Table S1. Try to sort the data and indicate the land use type: Forest, Cropland, Grassland. Based on this classification, you can then derive if the overestimation of soil C stock was prevalent in a certain land use type. Maybe one of the four calculation methods was used more frequently in a certain land use type? Check if this is feasible.

Response: We have done such a land use comparison and added another table in the supplement. We also added the following sentences: "Cropland was the land-use type in which stones were most often completely ignored. Eighty five percent of all reviewed cropland studies used M1 to calculate SOC stocks (Table S2). In contrast, 71% of all studies that used M4 were conducted in forest soils. This might be related to the fact,

that stones are more abundant in forest soils and that SOC investigations in cropland soils are often restricted to the surface layer. However, only 12% of all forest studies used method M4, while M1 was the most often applied (42 %)."

lines 107-108: why statistical analyses were not conducted? I'm not sure I understood, rephrase the sentence or explain in a different way. Or delete it if not pertinent with the rest of the manuscript.

Response: Statistics is used to separate random deviations from systematic deviations. If only systematic deviations occur, no statistics are needed. We changed the section, which now reads as follows: "Due to the fact that method-induced deviations were systematic, we did not conduct statistics. As soon as the stone content is not 0, there is always a significant difference between calculation methods, no matter how small the differences between methods would be."

TECHNICAL CORRECTIONS

*line 70: delete "where identified".

Response: We changed this accordingly.

*line 79: list the terms in the order they appear in the equation. i.e. SOC stocki, SOC con fine soil, etc.

Response: We changed the sentence as follows: "where ãĂŰBDãĂŮ_sample is the bulk density of the total sample, ãĂŰmassãĂŮ_sample is the total mass of the sample, ãĂŰvolumeãĂŮ_sample is the total volume of the sample, ãĂŰSOCstockãĂŮ_i is the SOC stock of the investigated soil layer (i) [Mg ha-1], ãĂŰSOCconãĂŮ_(fine soil) is the content of SOC in the fine soil [%] and ãĂŰdepthãĂŮ_i is the depth of the respective soil layer [cm]."

*lines 95-101: I would suggest not to repeat the equations which were already reported in the page before, but to cite them instead.

Response: We changed this accordingly.

*Figure 1: add a top x axis title "Volumetric stone classes"

Response: We added the title "Volumetric stone content classes".

---

## Author Comment (AC2) · 18 Jan 2017

We thank the reviewer for his time and effort he spent to read and comment on our manuscript and believe that numerous comments did help to improve the manuscript. We also explain why we disagree with some of his comments.

SPECIFIC COMMENTS Line 51: In line 44, you already mentioned that mineral fragments > 2mm contain pieces of the size classes "gravel" and "stone". But it is not clear how you use the term "stone content" in your manuscript. For example in the Guidelines for soil description of the FAO, the size class of "Coarse gravel" is between 2 and 6 cm and "stones" are classified as mineral pieces between 6 and 20 cm. Can you clarify this?

[Figure]

Response: We are referring to the "coarse soil" as a whole, so the fraction >2 mm. We agree that the introduction was not exactly clear on that and changed the sentence, which now reads as follows: "Coarse mineral fragments >2mm (gravel and stones, in the following only referred to as stones)..."

I confirm the comment of Referee #1, that the equations should not be repeated. Maybe they could be summarized in a table, which would allow to repeat the sub-equations in the columns (e.g. column 1: Method, C. 2: BD equation, C. 3: SOC stock equation)

Response: As suggested in this short comment and by referee #1 we decided to cite the equations where they were double mentioned before.

Line 70-71: Did these four authors which you cited develop the equations or did they only apply them? In the second case, please refer to the original reference. For example, Henkner et al. (2016) cited four publications concerning the SOC stock equation which was used by them. And during the description of the equations, could you refer to the original references in each section?

Response: The latter is the case. We were not able to find original references for each method. It is not possible, because sometimes the methods are also published differently as they are cited and referred to. But in any case, we do not think that it is important to find out who used which equation for the first time, but rather to present the existing equation ensembles and why they are not correct. We do not want to blame certain researchers directly for using incorrect equations.

Line 86: Do you have any references which underline that you use a stone density of 2.6? Or is it the average stone density of the dataset which you used?

Response: We added the following reference: Don et al. 2007.

Line 107: What was the criterion for the classification of the data of the stone content? How did you choose the boundaries of the classes?

Response: This classification is a compromise of not creating too many classes and to illustrate the effect of stone content sufficiently. It is not based on any existing classification scheme.

Lines 118-119: I do not see this observation in Table 1.

Response: This is correct. We referred to an old version of Table 1. We now deleted this reference.

Line 124: This sentence should be part of the methods. Is BD fine soil of 1.2 the average of the whole dataset (German Agricultural Soil Inventory)?

Response: No, this was an artificial, but realistic number. Figure 2 was actually decoupled from the German soil inventory dataset. However, we see that it actually makes much more sense to just use the average of the german inventory dataset, which was 1.4. We changed the graph and the respective sentences accordingly. However, to clarify that Figure 2 is only a theoretical example, we started a new paragraph to separate it from the German soil inventory results.

Line 174: Would it be difficult to recalculate SOC stocks which were done by other equations in former times by using Eq 7? Which modification of the raw data is needed? I ask you because such improvements may fail and do not find acceptance due to the bias which is produced by the change of the method. Therefore, it is the simplest way to apply a standardized and established method for dealing with time-series, for example, to keep the comparability at a highlevel. Could you therefore clarify how data of soil inventories like the inventory which you used must be re-organized first or could be used that the already taken samplesand available data is applicable with Eq. 7 in the future?

Response: It is only about the correct use of the stone content. If stone content is measured, stocks could be recalculated easily, even for published datasets, on which e.g. a resampling should be built on. We agree that it is good to mention this in the

manuscript and added the following sentence: "If stone contents were measured, also SOC stocks of existing datasets could be recalculated, e.g. in the case of resamplings."

Table 1: In my opinion you show two different topics in this table. Is it therefore meaningful to show them in one table?

Response: It is true that the topic differs. Both tables alone would be relatively small, so we thought combining both would save space. The complexity of the table content is limited, so we think we can leave it in one. Furthermore, the German dataset is only an example and we do not want it to get too much space in the manuscript and focus on this.

Table 1 and Figure 1: Is the data of Tab. 1 and Fig. 1 in a way comparable? In my opinion, the highest differences and hence the relative deviation of M4 is found in sampling depth 10-30 cm and not in 70-100cm (Figure 1). However, in Table 1 the highest value concerning the average relative deviation was found in 70-100 cm.

Response: Please study figure 1 again. It shows absolute carbon stocks for M1-M4. The highest relative deviations are found in 70-100cm, which becomes most obvious in the column of stones >30%.

Figure 1: Did you compute any significance test that you can show which method (M1-M3) differs significantly from M4 per depth and stone class?

Response: Statistic is done to separate systematic from random deviations. If there is only a systematic deviation, no statistic is needed. In other words: as soon as the stone content is not 0, there is always a significant difference, no matter how small the differences between methods would be. We changed the respective section, which now reads as follows: "Due to the fact that method-induced deviations were systematic, we did not conduct statistics. As soon as the stone content is not 0, there is always a significant difference between calculation methods, no matter how small the differences between methods would be."
Figure 1: Clarify what kind of values you show. Average values?

Response: We added the word "average" in the figure caption.

Figure 2: Would it be possible to show the 95%-confidence intervals of each of the functions? This would show the variability.

There is no variability. The figure shows what happens to the deviation in SOC stocks from M4 when the stone content increases. This relative deviation is independent of carbon concentrations. The only parameter that had to be added was bulk density of the fine soil, which was set to the value of 1.2. We now change that value to 1.4 (see comment above), which is representative for German agricultural soils.

Figure 2: How did you calculate the systematic deviation? Please, show the formulas at least in the Figure. Can you explain it in the method section? (see above comment line 124).

Response: We added the following sentence: "Therefore, we additionally calculated the method-induced potential deviation in SOC stocks as a function of stone content (0-70 vol. %) for the average ãĂŰBDãĂŮ_(fine soil) of the inventory dataset (1.4)."

Figure 2: What is the number of replicates per depth and stone class? Is the number of replicates reasonable for the different variabilities, to some extent?

Number of Observations <5% 5-10% 10-20% 20-30% >30% 1745 287 127 41 26 1625 314 143 70 94 1541 233 145 92 241 1512 211 143 92 283 1465 199 123 70 285

Response: We thought of including this table in the beginning, but decided that this information is not important for the message of the paper. Thus we derived the relative proportions of stones as included in table 1. As mentioned, the inventory dataset is only an example used to demonstrate the effect with "real data". Even very few observations in each class would be sufficient in this case since we talk about systematic deviations that are independent from the number of observations.

Further, could it be that samples of depth horizons 50-70 and 70-100 cm originate from deep soils like "Parabraunerde", "Böden auf Lössderivaten" or "Böden auf Sand", which can contain a low stone content in the upper sampling horizons and maybe also in the deeper sampling horizons? And soil from e.g the "Schwarzwald", "Mittelgebirge" and the Alps do not reach depths of 50, 70 or 100 cm. In these cases, the stone content is already high in the upper sampling layers like 0-10, 10-30 or 30-50 cm. Would this explain the higher variability in 10-30 cm besides the fact that the different depth intervals contributes also to the SOC stock (depthi)? Because during the single stratification by stone content, it seems that the different sampling horizons of all used test sites were mixed. Hence, the stone class (> 30 %) in depths 0-10, 10-30 and 30-50 is represented by sites fromrugged terrain while the depth classes 50-70 and 70-100 cm contain samples of "deep soils" which show a stone content > 30 % only deeper than 50 cm soil depth. How does this issue influence your results?

Response: It is correct, that not all depth increments are represented exactly by the same sites, due to the facts that some soils were not sampled to a depth of 100 cm. However, the main aim of this study is to show that the method of calculation has a major effect on the actual SOC stocks and that this effect increases with stone content. This is demonstrated using a real national soil inventory dataset. Since this dataset has different sampling depths, we analysed all of them to come up with average values for each depth increment and stone class. And as such they should be interpreted. They are country-wide averages and considered representative for Germany, which again is only used as an example.

Therefore, I agree with Referee #1 comment to re-classify the dataset to avoid such an assumed mixture. For this, the parameter "depth" in the inventory could be the classification factor. Or it could be possible to apply a spatial classification by soil type or region. This would be the first step. After this, the sub-datasets can be stratified by the stone content in a second step.

Response: We have classified the data by depth and stone content. Stratification into

land use types was tested (cropland vs. grassland), but the differences were marginal and not interesting to present: A grassland with 10% stones behaves like a cropland with 10% stones, a Schwarzwald-Luvisol with 10% stones behaves like a Podzol soil in the northern plain with 10% stones regarding the method-specific deviations. It is only the stone content that matters, so there is nothing to learn from further stratification. The message of this paper applies to all soils worldwide, may that be paddy soils, desert soils or high mountain soils. A stratification of Germany is thus useless for the aim of this paper. The comment of Referee1 did not refer to the German dataset but to the 100 reviewed publications.

Figure 3: Where do you refer to this figure. In my opinion, it should be Figure 1 and placed in the method section.

Response: We referred to this figure in line 140. Putting it into the method section was our initial thought as well, but since we only introduce the equations in M&M and discuss and evaluate them in section 3.2 only, we think that this is the right place for the figure. The figure needs some explanation and when this is done in M&M already, it is hard to avoid redundancy in 3.2.

Line 158: Do you mean SOC stocks instead of SC stocks?

Response: Yes. We changed that accordingly.

---

## Referee Comment (RC2) · B. van Wesemael (Referee) · 13 Feb 2017

The manuscript is clear and well-written. It addresses the bias in SOC stocks that could result from the correction for rock fragment content. This is an important topic and the paper could be well-cited. I am pleased to see that a detailed comment was already posted and that the authors replied to this comment as well as to the comments of reviewer 1. Thus, the majority of the minor errors and issues that were not clear are already dealt with. This short paper is valuable in correct estimation of SOC stocks and even allows a correction of available data bases. I have one major remark (see below line 159). I agree with the proposed use of FSSi (eq. 6), but I would not be surprised if many/some studies (out of the 36 ) using M3 already implicitly use this approach by

calculating the mass fraction of stones (see explanation below).

Minor comments: Line 28, 31 and throughout the manuscript: I agree with the comments posted on terminology of gravel and stone content. Already in the 1990's Poesen and Lavee (1994, Catena 23, 1-28) published a special volume on stony soils. I am not suggesting that you cite these authors. However, their use of the term 'rock fragments' avoids discussion on the size fraction of mineral particles > 2 mm, and I would recommend to use it. Nevertheless, the use of 'stony' as an adjective is fine for me. Line 47 See previous comment: 'coarse soil' is creating confusion, as commonly we think of the fine earth as soil. Could not you say 'the fraction > 2 mm' Line 70 Please reformulate in order to avoid using 'identified' twice in one sentence. Lines 81-82 I am not sure that I understand 'inadequate representation'. Method M1 overestimates SOC stocks as it does not correct for a volume of SOC free soil fraction i.e. the stones. Is this correct? Lines 96 and 101 I assume that the stone fraction is a volume fraction and not a mass fraction. Could you please specify this in the text?

Line 159: I would argue that M3 gives correct results if used with the mass fraction of stones instead of the volume fraction of stones. Writing the units of the SOC stock equation will hopefully convince you (not taking into account '%' for the concentration):

Stock = g C/g fine earth * g (fine +coarse) / cm$^3$ (fine +coarse) * cm * g fine / g (fine +coarse)

Simplifying this equation gives: g C/ cm$^3$ (fine +coarse) which is the stock. I believe that this approach is also frequently used in the literature, and maybe unfairly accounted for in your 36 studies in line 127. If I am not mistaken, the benefit of this equation is that you do not need the density of the stones. This approach is similar to your equation 6. After all, you also correct for the mass of the stones only.

---

## Author Comment (AC3) · 14 Feb 2017

Themanuscript is clear and well-written. It addresses the bias in SOCstocksthatcould result from the correction for rock fragment content. This is an important topic and the paper could be well-cited. I am pleased to see that a detailed comment was already posted and that the authors replied to this comment as well as to the comments of reviewer 1. Thus, the majority of the minor errors and issues that were not clear are alreadydealtwith. ThisshortpaperisvaluableincorrectestimationofSOCstocksand even allows a correction of available data bases. I have one major remark (see below line 159). I agree with the proposed use of FSSi (eq. 6), but I would not be surprised if many/some studies (out of the 36 ) using M3 already implicitly use this approach by

C1 calculating the mass fraction of stones (see explanation below). Minor comments: Line 28, 31 and throughout the manuscript: I agree with the comments posted on terminology of gravel and stone content. Alreadyinthe1990'sPoesen and Lavee (1994, Catena 23, 1-28) published a special volume on stony soils. I am not suggesting that you cite these authors. However, their use of the term 'rock fragments' avoids discussion on the size fraction of mineral particles > 2 mm, and I would recommend to use it. Nevertheless, the use of 'stony' as an adjective is fine for me.

Response: We changed stone content into rock fragments content in the entire manuscript.

Line 47 See previous comment: 'coarse soil' is creating confusion, as commonly we think of the fine earth as soil. Could not you say 'the fraction > 2 mm'

Response: We changed this into "fragments >2 mm".

Line 70 Please reformulate in order to avoid using 'identified' twice in one sentence.

Response: The second "identified" was deleted.

Lines 81-82 I am notsurethatIunderstand'inadequaterepresentation'. MethodM1overestimatesSOC stocks as it does not correct for a volume of SOC free soil fraction i.e. the stones. Is this correct?

Response: Yes. We changed the sentence as follows: "This method does not account for rock fragments at all". The introduced bias is discussed later and does not have to be mentioned here. Lines 96 and 101 I assume that the stone fraction is a volume fraction and not a mass fraction. Could you please specify this in the text? Response: Yes, that is correct. We now added [Vol. % /100] in the text.

Line 159: I would argue that M3 gives correct results if used with the mass fraction of stones instead of the volume fraction of stones. Writing the units of the SOC stock equation will hopefully convince you (not taking into account '%' for the concentration): Stock = g C/g fine earth * g (fine +coarse) / cm3 (fine +coarse) * cm * g fine

/ g (fine +coarse) Simplifyingthisequationgives: gC/cm3 (fine+coarse) which is the stock. I believe that this approach is also frequently used in the literature, and maybe unfairly accounted for in your 36 studies in line 127. If I am not mistaken, the benefit of this equation is that you do not need the density of the stones. This approach is similar to your equation 6. After all, you also correct for the mass of the stones only.

Response: After reconstructing the suggested equation, we agree that M3 is only wrong in the case that volumetric rock fragments fraction is used. We have now added the following sentence in the M&M section: "It has to be noted, that when the term rock fragments fraction in Eq. 5 corresponds to the mass fraction of rock fragments and not to the volume fraction, M3 resembles M4." Also we have revisited the 36 publications using M3 and changed the section as follows: "The literature review revealed that M1, M2, M3 and M4 were used by 52, 5, 30, and 13 studies respectively. In 19 out of 30 studies using M3, it was unclear if the correction term (1- rock fragments fraction) referred to the volumetric or gravimetric rock fragments fraction. Thus, in 68-87% of all studies reviewed, SOC stocks were systematically overestimated assuming a rock fragments fraction >0."

In addition, we also included the following: "For the probe method, the equation to calculate FSS_i can be further simplified to: FSS_i=mass_finesoil/Surface_sample (Eq. 8), where Surface_sample is the surface area [cm$^2$] of the sampling device. This might be of special interest for studies conducting sampling by fixed soil mass and not by fixed depth."